# Vice VRsa: Balancing Bystander's and VR user's Privacy through Awareness Cues Inside and Outside VR

Youngwook Do*
Georgia Institute of
Technology
Autodesk Research

Frederik Brudy†
Autodesk Research

George Fitzmaurice‡
Autodesk Research

Fraser Anderson§
Autodesk Research

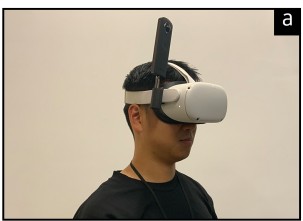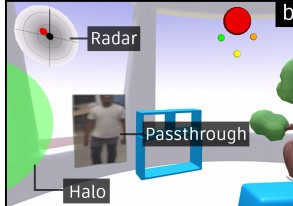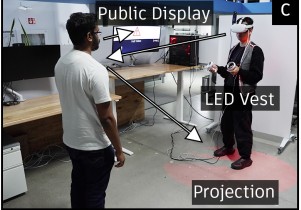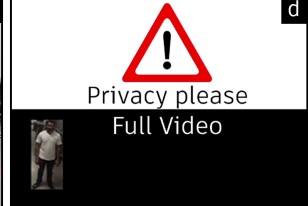

Figure 1: Due to the immersive VR experience, a VR user may not notice a bystander's presence, which subjects the VR user to being monitored by bystanders without knowledge. A VR user can use a VR headset's camera (a) to monitor their surroundings. However, conversely, this camera recording raises bystanders' privacy concerns as they may be recorded without consent. We introduce *Vice VRsa*, which is designed to balance VR users' and bystanders' privacy by providing awareness cues to (b) the VR user about a bystander's presence and location (Radar, Halo, Live View) and (c) to the bystander about a VR user's privacy mode and what is being recorded about them through a color display (projection and LED vest) and public display (c+d).

## ABSTRACT

The immersive experience of Virtual Reality (VR) disconnects VR users from their physical surroundings, subjecting them to surveillance from bystanders who could record conversations without consent. While recent research has sought to mitigate this risk (e.g., VR users can stream a live view of their surrounding area into VR), it does not address that bystanders are conversely being recorded by the VR stream without their knowledge. This creates a causality dilemma where the VR user's privacy-enhancing activities raise the bystander's privacy concerns. We introduce *Vice VRsa*, a system that provides awareness of bystander presence to VR users as well as a VR user's monitoring status to bystanders. This work seeks to provide a concept and set of interactions for considering mutual awareness and privacy for both VR users and bystanders. Results from preliminary interviews with VR experts suggest factors for privacy implications in designing VR interactions in public physical spaces.

**Index Terms:** Human-centered computing—Visualization—Visualization techniques—Treemaps; Human-centered computing—Visualization—Visualization design and evaluation methods

## 1 INTRODUCTION

Virtual reality (VR) provides users with immersive experiences in entirely virtual spaces. A VR user's immersion in the virtual space disengages their sense of presence from the physical space surrounding them. Such disengagement subjects VR users to not just running into a physical obstacle but also being monitored by others physically co-located without their knowledge or consent [34], as VR users may be unaware of their surroundings. This can result in putting the users in vulnerable positions in their physical space (e.g.,

*e-mail: youngwookdo@gatech.edu
†e-mail: frederik.brudy@autodesk.com
‡e-mail: george.fitzmaurice@autodesk.com
§e-mail: fraser.anderson@autodesk.com

private conversations being overheard or even recorded by someone co-located, accidental collision with physical obstacles, or other risks to their physical safety).

To alleviate such risks, VR users could activate the VR headset's passthrough camera to see their real surroundings. Additionally, researchers have previously explored various ways to make VR users aware of bystanders in their surroundings. For example, prior work demonstrated representations of the real world in the virtual environment by blending a camera feed with the virtual world [26, 51] or by bringing avatar representations of bystanders into VR [51].

However, these monitoring setups could in turn raise bystanders' privacy concerns as the passthrough camera is embedded in the headset to monitor a VR user's surroundings in a physical space. Using wearable cameras, such as those found in commercial VR headsets, remains a long-standing problem of unwanted surveillance [19, 46], as little awareness is provided to bystanders. To mitigate privacy concerns of bystanders in these VR hybrid settings (i.e., situations where a VR user is in a physical space where non-VR users might also be present), a camera activation indicator can be used. However, prior work suggests that a LED indicator may not be noticeable and understandable to end-users [6, 19, 48].

In this paper, we aim to level the playing field between VR users and bystanders, by providing awareness to a VR user about a bystander's presence (*VR user's awareness*) as well as providing bystanders with awareness about what a VR user might see about them (*bystander's awareness*). To that end, we present *Vice VRsa*, as an example of a broader concept of a system offering mutual awareness to VR users and bystanders about each other's monitoring status through inside- and outside-VR headset representations. Moreover, we designed *Vice VRsa* to allow both a VR user and bystander to negotiate their desired levels of privacy as the desired level of privacy is context-dependent [10]—a VR user may not care about being listened to during a casual chat in VR, but may be more mindful about who is around during a confidential meeting in VR. As a proof-of-concept, *Vice VRsa* provides a VR user with options to choose from four different modes to determine their desired level of privacy: none/green, low/yellow, medium/orange, high/red (See Figure 3). The VR user can change the mode to receive a different level of granularity of information about their surroundings, while the information they share about their activities inside VR decreases.

Concurrently, the bystander can be informed about the VR user's desired level of privacy through color indicators, as well as what the VR user is recording about the physical environment via an accompanying public display.

In summary, we contribute *Vice VRsa*, an instantiation of a concept that aims to improve both a VR user's and bystander's awareness of each other's monitoring status. Through our implementation, we demonstrate how *Vice VRsa* accommodates different privacy needs and how it allows bystander and VR users to negotiate their desired levels of privacy. Initial feedback on *Vice VRsa's* concept and system from expert VR users shows that the concept is easily understood and that experts find it promising to support their privacy needs in VR hybrid settings for both VR users and bystanders.

## 2 BACKGROUND AND RELATED WORK

The concept of *Vice VRsa* builds upon prior work from three areas: (1) VR users' privacy concerns against covert monitoring by bystanders; (2) balancing VR users' awareness about their bystander presence and bystanders' interruption; and (3) bystanders' privacy concerns against camera recordings without consent or knowledge. In the following subsections, we will outline our work's position in relation to prior work.

### 2.1 VR Users' Privacy Concerns against Bystanders in Public Settings

VR's immersive experience overrides the users' sense of presence in a physical space, putting them in a vulnerable position in terms of privacy. For instance, bystanders near the users could eavesdrop on their conversation without permission, or gain information by observing their interactions [25, 43, 49, 55]. Prior work also pointed out that a bystander could exploit a VR user's vulnerable state by recording video and/or audio of them without their knowledge or consent [31, 34]. Researchers have pointed to the need for an interaction that addresses privacy concerns for VR users in public spaces such as a shared office [30] as onlookers might still gain sensitive information from VR users' actions [16]. Consequently, researchers have explored how to prevent shoulder surfers from inferring VR users' data entry in VR, for example, by preventing bystander's observing or recording VR users' passcode-entry gestures with the hand-held controllers [25, 55] or when typing on their keyboard [43].

VR users could take off their VR headset [51] or activate the headset's passthrough camera to see outside. However, a VR headset's passthrough only shows a live camera feed from the headset's front-facing camera, with no option to see on the sides or behind, and removing a headset interrupts any task and breaks the immersion. Moreover, because the passthrough feed only provides a full-screen view, VR users must pause their activity to check bystanders' presence, even when requiring minimal information about their presence— VR users may just want to know if someone is nearby without recognizing who they are. To that end, prior work explores how to improve VR users' awareness of bystanders in proximity without breaking the immersion while helping them to be informed about their physical space. For example, researchers have demonstrated methods using various modalities: different visual cues such as radar views to get a quick overview of any bystanders' location, avatar as a way to represent precise bystanders' location information, passthrough videos to engage with the actual physical space outside VR application [13, 20, 26, 33, 45, 51]; auditory feedback as a means to enhance a user's auditory awareness about their physical space [13, 33], and text as unobtrusive representations [13, 26, 33].

In our work, we adapted and modified various representations of bystander presence inside VR. We, specifically, explore how such representations can be used in privacy-related contexts, and accommodate varying levels of privacy. Additionally, we consider a bystander's privacy against the VR device's camera recording.

### 2.2 Balancing the Disruption by and Awareness of Bystander Presence for VR Users

Interventions from the outside VR can disrupt the VR user's feeling of immersion. Particularly, a bystander's interruption of VR experiences increases a VR user's cognitive burden, and may even cause discomfort [26, 32, 37, 53]. George et al. found that a VR user is less likely to feel discomfort when they are interrupted from outside during their task switch (e.g., during the app transition) inside VR than in the middle of VR tasks [11]. However, Mai et al. found that not knowing information about their surrounding could cause cognitive burden [23] while putting the user at risk of bumping into objects such as furniture by accident or unwanted or abusive activity by the bystander [34]. Owing to that, there occurs a constant negotiation for users to choose between the needs of interruptions and focus.

In addition to bystanders' interruptions, how to represent bystanders in VR environments also affects the VR user's immersion. For instance, Kudo et al. explored three different representations of a bystander's presence inside VR [20]. Their findings show that an avatar representation of a bystander was most effective, although more peripheral visualizations of bystanders preserved a VR user's immersion better. They emphasized the need for systems to use the bystander representation that is most appropriate for the level of urgency a given task requires [20]. Yang et al. present ShareSpace which illustrates bystanders as virtual wall or obstacles and helps VR users to avoid physically bumping against the bystanders [54].

In order to handle the constant balance between immersion and interruption in VR, we build on prior work to create a concept and system that provides adjustable levels of awareness regarding bystander presence. *Vice VRsa* offers different bystander representations according to the level of VR users' desired privacy. We aim to give VR users agency over the granularity of the information they receive about bystanders' presence which is designed to match their situational privacy needs.

### 2.3 Bystander Privacy Concerns against Wearable Cameras

VR devices (e.g., a headset, controllers) have a multitude of sensors including built-in cameras and microphones, which enables detecting and observing bystanders without their knowledge. This poses a threat to bystander's privacy, as these sensors could unwittingly capture their directly identifiable (e.g., face) or otherwise personal information (e.g., private conversations), causing social friction [2, 17, 46, 48]. Transparency about the camera recording status can reduce this friction. Commodity wearable VR devices (e.g., Quest) or Augmented Reality (AR) glasses (e.g., Google Glass, Snap Spectacles) have an LED indicating to bystanders whether the camera is currently in use or not [14, 18]. However, such LED indicators are not easily noticeable and could even confuse bystanders or not be understood at all [19, 35]. To overcome this, researchers have sought ways to avoid undesirable camera capture and to transparently communicate to bystanders camera recording status [2, 48]. For example, Alharbi et al. found that the level of obfuscation of camera capture could affect the level of bystanders' privacy concerns against unwanted capture [2]. Also, *PrivacEye* demonstrated a way to improve noticeability and understandability by using a physical cover that blocks a camera lens when the camera is not in use [48].

Unlike prior work that addressed privacy concerns about AR, there has been little work regarding the privacy of a VR user's bystander. Schwind et al. found no evidence that bystanders of VR users have privacy concerns about being recorded. However, they also point out that privacy concerns with AR glasses only came about with increased popularity, which in turn led to the reduced social acceptability of these devices [44]. In our work, we extend prior work to consider bystander privacy for VR by providing awareness about what a VR user is recording about their physical space.

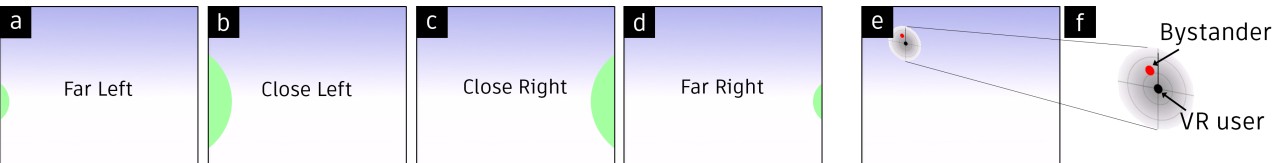

Figure 2: Halo implies information of (1) distance (2) and direction. (a, b, c, d) The bigger the sphere is, the closer a bystander is. The sphere appears on the left/right based on the bystander's position relative to the user. (e, f) The radar view shows the bystander's precise location in VR.

## 3 VICE VRSA

*Vice VRsa* is a framework and set of interactions to increase awareness of bystanders' presence to VR users and that of VR users' recording of their surroundings to bystanders to enable both sides in negotiating their privacy needs. Next, we will discuss the design considerations and interactions of *Vice VRsa*.

### 3.1 Design Considerations for *Vice VRsa*

We account for two factors to design *Vice VRsa*: (1) desired privacy depending on contexts and (2) privacy notice timing.

#### 3.1.1 Balancing Privacy and Awareness for VR Users and Bystanders

***Inside VR Representations for VR Users' Privacy against Bystanders.*** People have different levels of desired privacy depending on their context and current situation [10,29]. For example, when a VR user performs an authentication gesture to log in to a VR application and does not want someone to watch their gestures, they may want to check outside to see if anyone is around them. Conversely, when they play a game and do not mind if someone is watching, they may not necessarily want to see outside their virtual environment to not break their immersion. Therefore, a system should offer different levels of awareness according to the desired levels of privacy, which echoes prior work's findings [12,27].

***Outside VR Representations for Bystanders' Privacy against VR Users' Monitoring.*** There is a longstanding problem of cameras in public spaces, with people expressing concerns about being recorded without knowledge or consent [6,48]. Even though commodity VR/AR wearables have an LED indicator associated with their camera activation, prior work found that the LED indicator is unnoticeable and confusing to end users [19,35]. One way to address such concerns is to build trust [1,7] by improving awareness of the camera's recording status [9]. This could, in turn, help people take further action if they did not want to be recorded. However, commercial VR headsets contain a multitude of cameras, allowing the VR user to observe or record their physical surroundings. This will only increase with future generations of hardware. To mediate trust between the bystander and the VR user, the system should therefore provide awareness to bystanders of whether and what the VR user is recording, as well as an awareness of the VR user's activities inside VR, allowing bystanders to regulate their behavior to accommodate the user's desired level of privacy.

#### 3.1.2 Timely Notices for Privacy Awareness

Understanding when and how personal information is collected helps people to protect their private data and avoid sensitive data to be tracked, monitored, or recorded without their knowledge [5, 21, 41]. As a result, privacy notices have become an essential part of interactive system as the privacy notice informs users about the data collection, allowing them to make their own privacy decision [5]. Researchers have indicated 'timing' as an important factor of the privacy notice to lower their oversight and to promote transparent communication on the data collection [3, 40, 42]. For example, a privacy notice that appears *during* smartphone app use is less likely to be overlooked than a privacy notice displayed at the app install

time [3]. Therefore, it is critical to find a 'right' moment and duration when a user stays informed about the data collection.

This motivates us to design *Vice VRsa* to provide a privacy notice *during* VR use to both the user and bystanders. The notice level and amount of information shown depend on the amount of information that is being recorded. For example, if the VR user only wants to know about the presence of any bystanders we show a lesser notice to the bystander. If however, the VR user wants to see a full video feed of any bystanders, the notice carries more urgency and details the information that is being recorded about the environment. We show notices simultaneously in and outside VR *during* the monitoring, providing awareness to both the VR user and bystanders so they can negotiate their desired level of privacy.

### 3.2 *Vice VRsa* Interactions

We showcase a set of interactions of *Vice VRsa* aligned with our design considerations. The system setup comprises two parts: (1) a representation inside VR that provides awareness about bystanders according to the user's desired level of privacy (Figure 3, left column); and (2) representations outside VR, including (a) color mode indicators showing the user's set privacy level and (b) an accompanying public display showing information about the VR user's activity and the information that is currently being recorded about the surroundings (Figure 3, right column).

*Vice VRsa* operates in four privacy modes, each corresponding to the desired level of a VR user's privacy. In each mode, various awareness cues in and outside VR are used, which act as privacy notice *during* VR use. We will discuss how to define the modes and the representations to use for each mode in the following subsections.

While various modalities (e.g., visual, sounds [31]) could be applied, we focus on visual feedback as an example of demonstrating how to design representations both in and outside VR.

#### 3.2.1 Awareness Cues Inside VR

Inside VR, we chose the representations depending on the extent to which they convey the information about bystander presence: (a) a *Halo* indicating bystander's presence and distance; (b) *Radar* showing a bystander's position on a radar map; and (c) *Live view* showing a live camera view of a bystander at their position in the physical space. Each of these views shows an increasing amount of information about the bystander, therefore also needing to record more information about the surroundings. We discuss the implementation of location and distance recognition in Section 3.4.

**Halo.** We chose Halo as a way to provide minimal information about bystanders' presence while keeping the disruption to VR immersion low. Prior work shows that representations with primitive designs (e.g., sphere) for bystanders provide minimal interruption to end-users [15]. To that end, we adopted the Halo approach which indicates the location of off-screen objects on a map application [4]. We designed the sphere representations (Figure 2a-d) to appear on the side, in the direction where a bystander is detected, to give a rough indication of presence, direction, and distance. Specifically, the size of the sphere indicates the distance of the bystanders from the VR user, and the location of the sphere indicates the general direction (left or right) of the bystanders. For this view, the system tracks a rough distance (close, medium, far) and position (left, right)

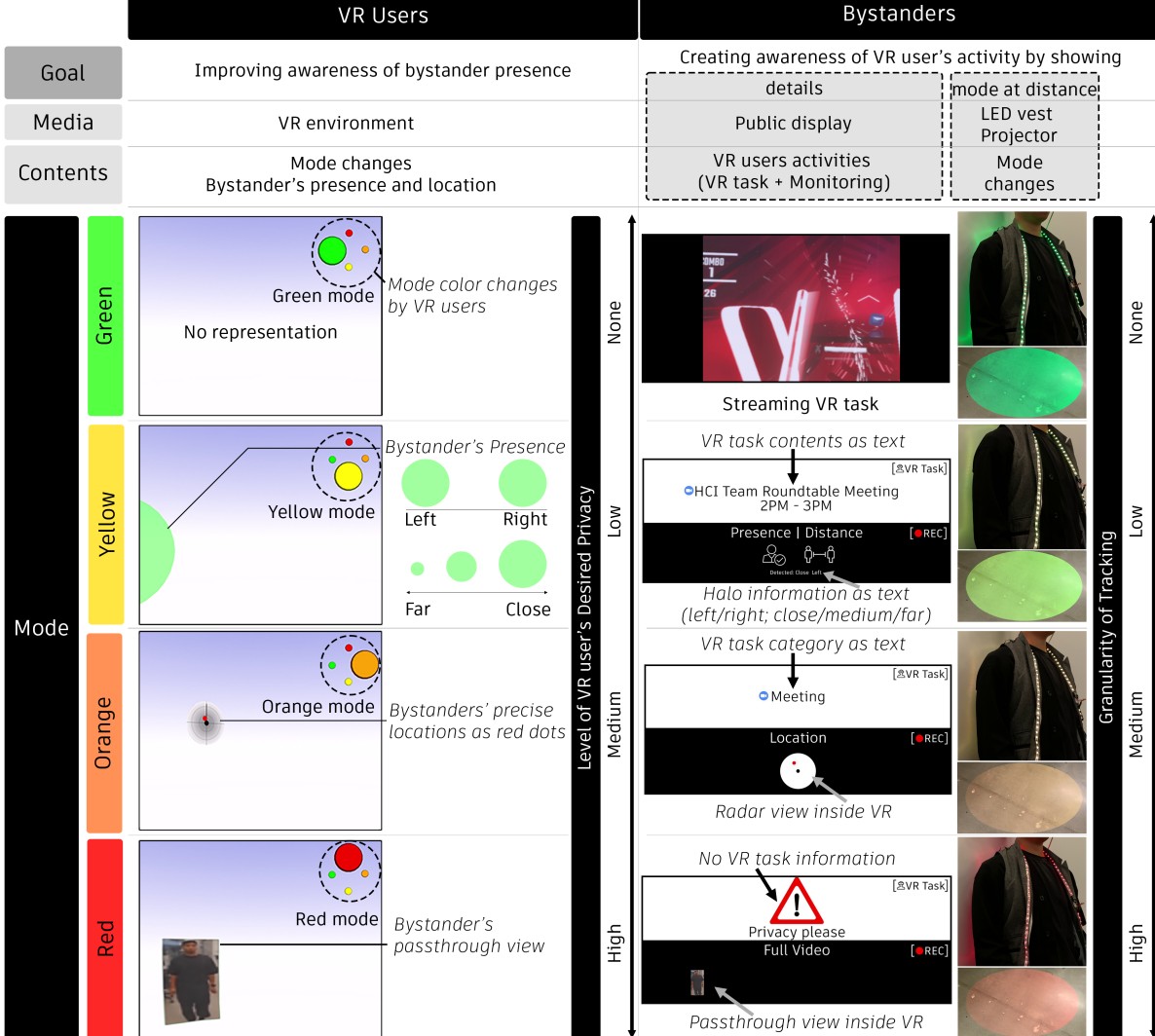

Figure 3: *Vice VRsa's* interactions can consider two stakeholders: (1) VR users; and (2) bystanders. Depending on the VR user's privacy mode settings (Green, Yellow, Orange, Red), representations both in and outside VR change accordingly. Further implementation details are provided in Section 3.4.

of the bystander. The Halo shows a rough estimate of bystanders' direction and distance. For more fine-grained information about bystanders' whereabouts, VR users can use *Radar* or *Live View*.

**Radar.** We chose Radar as a means to offer a quick overview of bystanders' presence for VR users to quickly examine more than the minimal information about bystanders (e.g., the number of bystanders, and how far they are) while maintaining the immersion. Kudo et al. presented the radar view as a way to display an "overview" of bystander locations without breaking the immersion significantly [20]. We adopted this representation and showed it in the top-left corner of the VR user's view (See Figure 2e). Red dots represent bystanders' precise locations relative to the VR user's position (See Figure 2f).

**Live View.** As a tool to see detailed information about bystanders' presence, we designed 'Live View' that displays the camera view of bystanders. In this case, the VR user prioritizes acquiring information about bystanders over maintaining a high level of immersion in VR, e.g., due to highly sensitive tasks or an increased possibility of strangers in the vicinity. Willich et al. demonstrate displaying a passthrough video of passersby in the VR view by using a Microsoft Kinect V2 depth sensor [51]. To achieve a similar effect, we use a 360-degree camera and stream a cropped video view of the detected bystanders. Our aim with this work was to demonstrate the concepts of *Vice VRsa*, rather than a production-ready implementation. Therefore, using a 360-degree camera offers two benefits over a VR headset's built-in passthrough camera. First, it allows users to see the full surrounding area, unlike front-facing recording by the passthrough camera. Second, the built-in passthrough feature does not provide API access to the raw camera feed. We, therefore, used a 360-degree camera to do image processing on the camera feed. The 360-degree camera and the headset's position are physically aligned through a custom 3D printed holder, therefore the camera feed appears as if directly recorded through the head-mounted display's (HMD) cameras. For this view, the system not only tracks a bystander's location but also records the live camera image.

### 3.2.2 Awareness Cues Outside VR

Outside VR, the visual cues indicate to the bystander what information the VR system is recording about them. The representations comprise (a) a *color mode indicator* showing the VR user's desired level of privacy; and (b) an accompanying public display (Figure 3, right column) providing details on the user's activity and what is

being recorded about their surroundings. Similar to inside-VR representations, the degree of information about bystander presence varies both in the color indicator and the public display.

*Color Indicator.* A VR user's cameras could record bystanders even over a distance, as long as there is a line of sight. The color mode indicators aim to provide an awareness of the VR users' selected privacy (and thus recording) at a distance within line of sight. A projection on the floor, utilizing a projector mounted above the VR "play" area using a tripod, is a direct indication of the user's selected privacy mode and is in direct proximity of the VR user whom it concerns (Figure 4d). An LED-enabled vest (Figure 4e) is aiding this by making the mode visible at even further distances. The floor projection and LED vest could also be replaced or aided by additional indicator lights directly on the HMD, and are used interchangeably.

*Public Display.* The public display is set up near the VR user's "play" area and visible to passersby (Figure 4a). The public display is split into two parts: the top half shows details on the activity the VR user is doing; and the bottom half shows what the system is recording about the user's surroundings (Figure 3 right column). The contents of both parts are controlled by the VR users' desired privacy level, as described next.

## 3.3 Color Modes Indicating Desired Levels of Privacy

The VR user can set their desired level of privacy by choosing between no (green), low (yellow), medium (orange), and high (red) privacy as shown in Figure 3. Each mode results in a combination of the previously described awareness cues inside VR, and corresponding visuals outside VR (showing a VR user's activity and what is being recorded). In this section, we describe the representations and interactions in and outside VR for each of the four modes. **Note:** The modes defined here represent example settings, but we anticipate different users could configure the behavior of their own privacy settings to best match their working contexts, such as the work they conduct, the physical space they are in, the people they share the space with, and their subjective privacy perception.

### 3.3.1 Green Mode: No Privacy

The green mode is designed for the context where a VR user does not need any privacy, and wants to minimize distractions from any awareness cues, for example, when playing a game.

*Inside VR.* In the green mode, no awareness cues about bystanders are provided inside VR (Figure 3, left column, "Green" mode). The system does not track any information about bystanders.

*Outside VR.* Because there is no privacy desired, the VR user's full VR content is streamed to the public display, allowing bystanders to see what the VR user sees (Figure 3 right column). The color indicators (projector and LED vest illuminate green) indicate the user's low privacy mode at a distance.

### 3.3.2 Yellow Mode: Low Level of Privacy

The yellow mode is for situations where the VR user needs a low level of privacy, yet wants to maintain a general awareness of bystanders' presence. The VR user's yellow mode aims to provide an awareness of general presence and proximity of bystanders.

*Inside VR.* The Halo appears on the side(s) of the screen the bystander's location in relation to the user. Its size implies the bystander's distance: the larger the circles the closer bystanders are.

*Outside VR.* The color indicators change to yellow, which signals to bystanders that minimal information about their location is being monitored by the VR user. The public display (Figure 3 right column) does not show the full VR content anymore but instead shows descriptive details on the VR user's activity, for example, the name of the application they are using, or if they are in a meeting,

the meeting invite's title and duration. The bottom part of the display shows what kind of information the system records about the environment: the bystander's rough direction and distance.

### 3.3.3 Orange Mode: Medium Level of Privacy.

The orange mode is intended for situations where the VR user is in need of a medium level of privacy. For example, they do not mind if bystanders know that they are in a meeting, but do not want any details known about it.

*Inside VR.* In the orange mode, the VR user receives more details about bystanders' location in the form of the radar view. This provides them with more precision on bystanders' locations and distances. In addition, the Halo is also shown.

*Outside VR.* The color indicators change to orange, which informs bystanders that more information about their presence is being observed. The public display's top half shows a general notice about the type of activity the VR user is doing, for example, that they are in a meeting or playing a game without revealing which one. The bottom half shows a notice that the bystanders' location is being recorded as well as the radar view's duplicate, similar to inside VR.

### 3.3.4 Red Mode: High Level Privacy Needed.

The red mode represents the highest level of privacy, where the VR user wants to be aware of bystanders and does not want any information about the VR tasks to be revealed.

*Inside VR.* As the information in the VR tasks is sensitive, the VR user needs to check *who* the bystanders in proximity are, to make an informed decision on whether it is safe to continue their activity or whether they should be mindful of their conversations and actions. Therefore, in this mode, the previously described live view is shown, which provides a window into the real world. Since this is in addition to the Halo and Radar View, the VR user is first made aware of a bystander's presence and location through these. Once aware, the user can turn their head in the direction the Halo/Radar indicate to see look into the real world using the live view.

*Outside VR.* The color indicators change to red and the public display shows a warning sign on the top part with a request for privacy. The screen's bottom half shows the live view of the 360 camera, where bystanders can see themselves (Figure 3 right column). This enables alerting the bystanders that the VR user needs a high level of privacy, and that the system actively records the bystander.

### 3.4 *Vice VRsa* Implementation

The *Vice VRsa* prototype, consists of a VR headset with an attached 360-degree camera for tracking bystanders, as well as a privacy notice (in our implementation an external display) and privacy awareness indicators (here an LED vest and an overhead projector).

### 3.4.1 Inside VR Representation

For the VR user, there are two hardware components: the VR headset and an external 360-degree camera which is used to recognize and track bystander presence. We use a Meta Quest 2 [28] and Ricoh Theta S 360-degree camera [38] for the VR headset and the 360-degree camera respectively. Current VR headsets allow users to activate a passthrough and to see a live view of their surroundings. However, the passthrough view is only either on or off entirely (as discussed in section 2.1), and the APIs disallow third-party developers to access the passthrough image. We, therefore, mounted the 360 camera on top of the VR headset to synchronize the head and the camera orientations, using a custom 3D-printed bracket (see Figure 4b). This allowed us to use image processing on a real-world view for bystander detection. The camera and VR headset are tethered to a PC via USB. To detect bystanders outside the VR environment, the PC receives the live video feed from the camera and processes it using the computer vision algorithm Yolo [36] running on Processing.

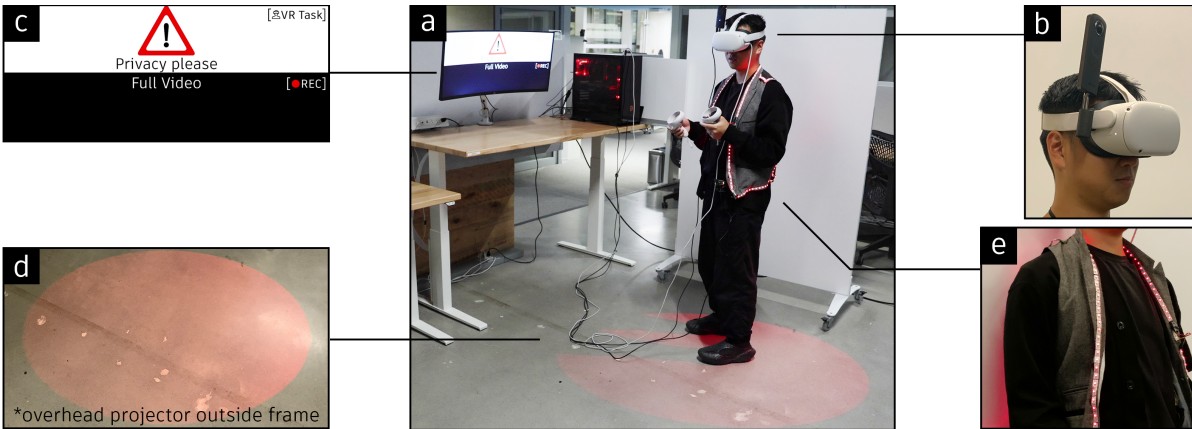

Figure 4: Vice VRsa setup (a) consists of four components: (b) Meta Quest 2 with Ricoh Theta S mounted on top; (c) public display; (d) projection on the floor; and (e) a vest attached with a series of LED strips.

Depending on the VR user's privacy setting, representations of the bystanders change inside VR. We created the representation in a virtual environment using Unity. Bystanders' location data is sent from Processing to Unity via JSON communication and the 360-degree video is streamed directly from the camera feed. The location data of detected bystanders consists of two types of information: (1) the angle difference from a user's current orientation; and (2) the distance. For our prototype, as a proof-of-concept, we estimate the distance of a bystander by the bottom pixel's y-axis coordinate of the bounding box of the bystander's body, assuming that the further away a bystander is located, the higher the bottom pixel y-axis position. We use this estimate of a bystander's distance and position (in relation to the VR user) for all the representations that visualize the distance and location of bystanders. For our proof-of-concept, we estimate the distance based on pixel location in the camera feed. We optimized our code for the office lab space where we deployed the system. As a rough guide, we consider distances of <3 meters to be *close*, >3 meters and <7 meters to be *medium*, and >7 meters to be *far* away. In the Unity application, this bystander data is used to display the system's awareness cues.

### 3.4.2 Setup for Outside VR Representation

The setup for outside VR representation consists of two components: a public display and color mode indicators (comprising a vest with LEDs and a projector). For the *Color Indicators* outside VR, we use a projector mounted on a tripod, connected to the PC via HDMI. The projector is positioned above the VR user's "play" area and projects onto the floor. The LED vest has an RGB LED strip woven into the garment, which is controlled by an Arduino microcontroller, connected to the PC, and controlled via Serial communication.

## 4 SCENARIO

To illustrate how *Vice VRsa* can support bidirectional awareness of VR users and bystanders, we describe the following scenario where Victoria (a VR user) and Bob (a bystander) working for different companies, while physically co-located in a co-working space.

### 4.1 Victoria's Perspective: VR User

***Green Mode*** Victoria wears a VR headset and plays a game for a break during work. While gaming, she does not care if someone is watching her. In fact, she wants to encourage others to join her in playing the game. Thus, she sets *Vice VRsa's* privacy mode to the green mode, and no representation of any bystander is displayed inside her VR space (Figure 5a).

***Yellow Mode*** After finishing the game, she joins her team's weekly social meeting through VR and starts a casual conversation.

This meeting is casual and nothing sensitive is discussed. While engaging in the conversations, she wants to know if someone is around in the physical space where she is located. She is worried that talking or laughing loudly might disturb people around her. She therefore adjusts the privacy mode to the yellow mode which allows her to be informed about bystanders' presence. A large Halo appears on the left-hand side, telling her that there is someone close by. Thus, she decides to keep her voice down (Figure 5b).

***Orange Mode*** After the team social meeting, she joins her team meeting where several important items are discussed. She not only wants to be mindful of disturbing others but also to have a thorough understanding of her surroundings (e.g., how many people are nearby and where are they). She changes her privacy mode to the orange mode, which activates the radar view. She can see there is one bystander on her left close by. While she feels okay that there is someone nearby, she chooses her words carefully to not disclose too much information about the discussion items (Figure 5c).

***Red Mode*** After her team meeting, she needs to have a confidential meeting with two team members about a new car design they have been working on. Due to the sensitive nature of the information, she wants to make sure no unauthorized person gains any knowledge about the confidential information. She, therefore, switches to the red mode, which not only provides her the awareness about bystanders' presence and distance through the Halo and Radar but also allows her to see outside the VR through the live view whenever there is a bystander. After a Halo appears on her left, she turns her head and sees a stranger nearby. She politely requests privacy, which the bystander adheres to. Victoria can now be certain that no one is around and continue her confidential design review (See Figure 5d).

### 4.2 Bob's Perspective: Bystander

***Green Mode*** When Bob is walking around the office, he finds that Victoria is wearing a VR headset and vigorously throwing her arms around. He sees her being engulfed in green light. He understands this as an invitation to come closer. He stops closer by and watches Victoria play a game in VR, which is being relayed on the nearby public display (Figure 5e).

***Yellow Mode*** After a few moments, Bob notices that Victoria sat down to have a conversation. He sees the lights around her change to yellow and that the public display shows Victoria having her weekly team social meeting which he believes is not sensitive. He also sees on the display that his rough location seems detected as 'close left'. While he is now aware of the change and the system recording information, he does not feel that he needs to leave (See Figure 5f).

***Orange Mode*** A few minutes later, Bob notices that the colored lights changed to orange. He feels more cautious and hears Victoria

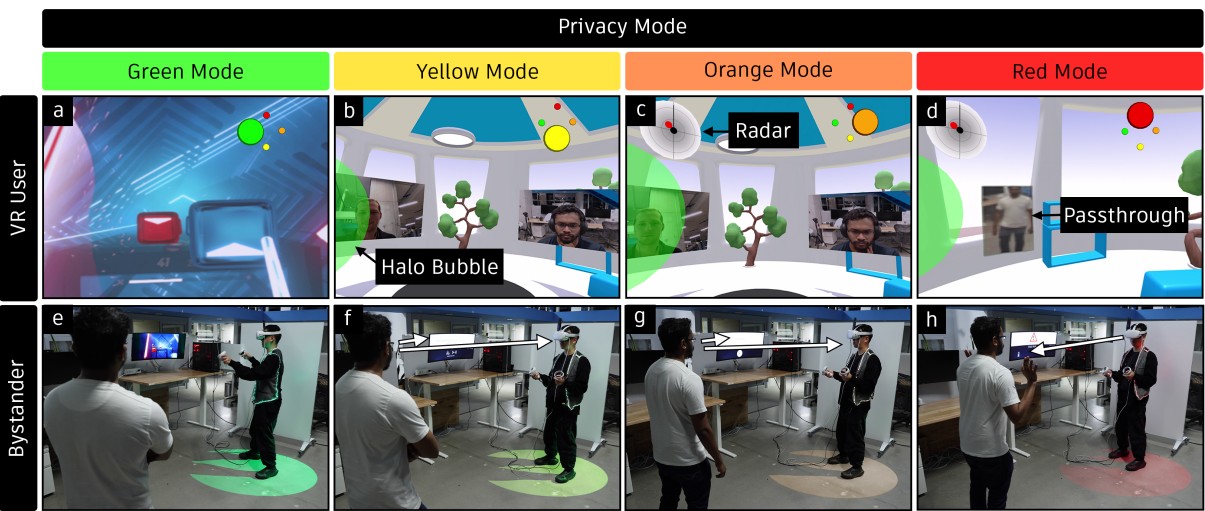

Figure 5: We illustrate a scenario to depict how Vice VRsa's interactions can be used based on the level of privacy mode in various contexts of a VR user (a, b, c, d) and a bystander (e, f, g, h).

talk about work topics. Bob notices that the public display shows his precise location on a radar view in relation to Victoria's position, and he sees that she is no longer in the team social meeting but cannot see any meeting details. He feels that he should not be close and overhear her conversation. Thus, he takes a few steps back from her and continues working on his phone (Figure 5g).

**Red Mode** Suddenly, Bob notices that the lights around Victoria turn red. Right after that, Victoria turns her head towards him. She speaks to him and asks him for privacy. Additionally, Bob finds that the public display shows a warning sign and message underneath that says 'Privacy Please!'. Therefore, Bob understands that she needs complete privacy and walks away immediately, giving Victoria the requested space (Figure 5h).

## 5 INITIAL SUBJECTIVE FEEDBACK

Our focus of this work is to evaluate a newly introduced concept of the system rather than assess the system's usability. Thus, we wanted to gain initial insights and feedback on *Vice VRsa's* utility, usefulness through interviews with expert VR users. The goal was to learn how *Vice VRsa* could increase awareness about bystanders, if sharing what is being recorded about the environment could be useful for bystanders, and how *Vice VRsa's* components would be understood. The study underwent our institution's internal ethics review process. We recruited expert VR users via email from within our institution to participate in a guided walk-through of the functionality of *Vice VRsa* as demonstrated through video scenarios. Sessions lasted one hour and were conducted via Zoom. Participants were compensated with an equivalent of 75 USD.

We recruited seven participants (1 female, 6 male; 29-53 years old, M=38.5, SD=6.97) with an average of 3.7 years experience of professionally working with VR (1-5 years, SD=1.39). Except one, all participants were currently working weekly or daily in VR, and their roles ranged from XR product, UX, and instructional content designers, to XR customer success managers, and XR researchers. Their professional experience with VR ranged from developing concept designs, training people or demonstrating VR applications, to using VR systems for architectural design tasks.

### 5.1 Study protocol

After obtaining consent and answering any open questions from participants, we first collected basic demographic information and asked them about prior experience conducting professional work in VR, such as length of experience, physical location, and type

of work. We then walked them through two scenarios. For the first scenario, participants were asked to imagine they are a VR user, working in an open-plan office with other people nearby. We then showed them video clips of how they might be overheard by bystanders (similar to the problem statement shown at the beginning of our video figure), followed by interview questions about how relatable such situations are, how they assert their privacy needs in such situations, and whether they have experience conducting privacy-sensitive tasks in VR.

For the second scenario, we asked participants again to imagine themselves as the VR user and showed them videos of the bystanders as well as the VR user's perspective of each privacy mode of *Vice VRsa* (green, yellow, orange, red), stopping at several points throughout the videos to ask questions. After presenting each visual indicator, but before explaining the meaning, we asked participants to explain their understanding of its meaning and how they would make use of this information. We then went on to explain the visual representation and asked for feedback on the perceived usefulness of the technique and any further feedback the participant may have. After each technique, we asked participants to reflect on whether the situation shown is relatable and their personal experience of similar situations. After presenting all the techniques, we conducted a final semi-structured interview and collected subjective feedback on the distraction and usefulness of the overall system.

### 5.2 Bystander's perspective

From the bystander's perspective, all participants agreed that the awareness of the VR user's desired level of privacy was useful to regulate their behavior (i.e., by seeing the color indicators from a distance and the details on the public display closeup). However, while all participants stated that green and red would be universally understood, they also stated that the color coding would need to be learned before fully understood. All participants agreed that getting information about the VR user's activity was helpful, primarily to know if they could be interrupted and, in the case of the green mode, finding opportune moments to do so. P7 compared this to regular desktop usage in a shared office, where "you can see what everyone is doing and if they're on a call" (P7). Interestingly, six participants also stated that the more privacy-sensitive a task is, the less likely they would interrupt the VR user, and this is also reflected in their behavior toward interrupting VR users. While all agreed that they would regulate their behavior depending on the VR user's privacy settings, 4/7 also stated that the system would intrigue them

to watch the VR user. This points to an interesting dilemma, where the addition of color indication and public display could create a honeypot effect, opposing the purpose of the system.

### 5.2.1 Awareness about Unwanted Recordings

Only 3/7 participants stated it was useful as bystanders to know *what* was being recorded about them. In fact, most would not mind having their distance or location tracked in general, but as soon as their video feed would appear inside VR, they would want to know about this and find the system useful. As one participant put it "[As a bystander] I don't care about position, but if [the VR users] are watching me, they are intruding my own privacy. Especially when recording my face, this would be going a little too far."

In sum, participants valued the information on the public display for awareness about the VR user's activity as bystanders, to know when to regulate their presence and opportune moments for interruptions. While stating that the display visuals would pose a learning curve, they mentioned that a few aspects of the system could be easily understood, especially in the most restrictive (privacy warning sign) and most permissive (VR content live streaming) modes.

### 5.3 VR user's perspective

From the VR user perspective, 6/7 participants would find *Vice VRsa* useful for awareness of their surroundings and 5/7 could imagine using the system. Several participants found *Vice VRsa* unnecessary at this stage as they mostly work from home and would hear others enter their office. However, all agreed that when working in a shared office or public space, bringing situational awareness into VR and sharing one's privacy needs with bystanders would be useful.

### 5.3.1 Awareness about Unwitting Monitoring

While most participants stated that their usage of VR in a public/shared space is usually for non-confidential tasks (e.g., trade shows, or playing games), they said that for any privacy-sensitive tasks, they would move out of the public space (e.g., by going into a closed meeting room). Most agreed that they would continue doing so for highly sensitive tasks (such as performance conversations (P4), client meetings (P5), or confidential design reviews (P2, P1)). On the contrary, P3 stated that even while working on sensitive prototypes, they would not always move to separate space, as "[bystanders] can't see what I'm seeing and cannot really make out what I'm talking about [from the] fragments they catch". However, several participants (P4, P7) also stated they are self-conscious about their actions when using VR and would therefore find the awareness about bystanders useful to "[not] look silly" (P7).

Using *Vice VRsa's* red mode to identify who is around was highlighted as most useful. This allows VR users to decide if they would need to break out of VR to negotiate their privacy needs or if they could ignore the bystander. P3 even pointed out that it would be good if the system could automatically identify if the bystander was trusted and not show any awareness cues if so.

### 5.3.2 Distraction / Breaking Immersion

Four participants mentioned that they would find the visuals distracting, especially the green halo as it grows directly in their peripheral vision. The radar view, however, was seen as less distracting as it "just sits there in the corner and I can ignore it [. . . and] the red dot doesn't suddenly grow into my face" (P6). While participants agreed that setting a VR user's privacy mode would be useful to communicate with bystanders, several participants also were unsure if the yellow and orange modes were needed. They found the green mode useful to communicate their actions to bystanders for the purpose of sharing the experience (e.g., gaming or teaching) and creating an awareness of what they were doing so that bystanders could adjust their behavior (e.g., they knew if the VR user is interruptible and if

so, identifying an opportune moment for doing so). The red mode was seen as useful to clearly express the need for privacy.

## 6 DISCUSSION

We explored a variety of exemplary interactions designed to balance VR user and bystander privacy and awareness through the implementation and evaluation of *Vice VRsa*, which includes the development of several novel interactions to improve mutual awareness.

### 6.1 Necessity for Privacy

One interesting finding from the expert feedback was about the necessity of the system from the VR users' perspective. Several participants mentioned that if they were truly doing something private, they are either already in a private space (e.g., a home), or would move from a public space to a closed meeting room. However, it is not always possible to move to a meeting room due to the availability, or the portability of necessary hardware (motion trackers, desktop computers, etc), especially for higher-end VR setups. Additionally, many of the participants have largely been working from home in recent years, which may impact their sensitivity about privacy.

Similarly, not all of the experts interviewed considered bystander awareness about the tracking completely necessary as they had no problem with the VR user knowing their presence. However, when they became aware that the VR headset cameras could record them without their knowledge, their perspective shifted to a more privacy-minded approach. It is foreseen that the use of these devices becomes more commonplace as with AR devices [47]. The sensors become higher fidelity and the real-world privacy risks associated with their use will increase. If not carefully designed, VR hardware could be used to record individuals without their knowledge and erode trust between VR users and bystanders.

### 6.2 Awareness vs. Intrusion

In our work, while the goal was to examine that the concepts were clear and that participants could understand the prototype's functionalities, several participants commented on the representations' implementation. For example, the halo was too large and distracting whereas the radar was seen as less-intrusive. We believe that further refinements could be made to these techniques in practice. For instance, shrinking the size of the halo or increasing its transparency could reduce the interruption to the main task.

Beyond visual feedback, other modalities could be considered as a way to be less intrusive both for the VR user and the bystander in the environment. For the user in VR, subtle haptic cues may be used to indicate the direction and presence of a person, or the use of air flow could provide the simulated sensation of a person walking by [39]. Spatial audio could perform a similar function, supplying information about bystanders through a potentially unused modality [33]. For the bystander, directional speakers may alert them as they walk into the area where they may be sensed, and cause them to look around for further cues about what might record them.

### 6.3 A Privacy Arms Race

During our feedback sessions, several expert VR users pointed out that the system is creating an additional overhead, by relying on the user manually setting the privacy mode. While, in this paper, we aimed to explore *Vice VRsa's* various exemplary interactions, future work should explore how a privacy mode can be inferred from the user's activity—e.g., a system could know that meetings are generally private. A system could go even further, identify a bystander and automatically determine if they are privy to the meeting's content. However, this creates an even bigger dilemma where the system not only records the video footage of the bystander but also connects their identity and contexts from additional data sources. This presents another embodiment of the ever-present arms race between defenders of privacy and those who exploit information.

## 6.4 Social Protocols Still Mediate Interactions

Our preliminary evaluation suggests that *Vice VRsa* can add value to VR/bystander interactions and can be a tool to help support interpersonal communication in the face of a technological barrier. However, the interfaces explored with *Vice VRsa* still rely on humans to follow established social protocols and need to be further studied to handle more intricate real-life scenarios. For example, the awareness indicators for both bystanders and VR users are designed to replace or augment some of the natural perceptual and social cues that are lost when users enter VR. However, the color code of the privacy modes needs to be communicated to and understood by both bystanders and VR users to use the system as intended. Even if both groups are aware of the color codes, *Vice VRsa* is not intended to prevent bad actors on either side. Furthermore, it is possible that bystanders may exploit the red privacy mode for intentional eavesdropping, as it indicates that a conversation might be worth overhearing. Therefore, for future work, it would be essential to further explore the broader design space of more complex circumstances and nuances in the real world.

## 6.5 Ecological Validity

Our focus of the initial study was to run a preliminary evaluation of a newly introduced concept rather than evaluate the system's usability. To that end, our study goal was to demonstrate the system's workflow and illustrate the system as Ledo et al. suggested as a 'demonstration' [22] and gather feedback on the interaction concepts' potential usefulness. Accordingly, we conducted our evaluation with an interview study by showing the videos that describe *Vice VRsa* interactions. While participants could not directly interact with *Vice VRsa*, we found that the concepts of *Vice VRsa* were understood and that participants felt that *Vice VRsa* could help improve the balance between bystanders' and VR users' privacy.

Furthermore, we chose to gather feedback from VR experts to understand whether and how any newly introduced privacy-related interactions could potentially jeopardize the overall usage of VR. Since privacy is a secondary concern for regular end-users [8], complicated interactions may discourage their adoption [52]. In the future, it will be interesting to collaborate with privacy experts to assess and refine the interaction design and to deploy *Vice VRsa* in the wild to run a longitudinal study to examine its long-run effect. This will help close the gap between in-lab and field studies [24]. For example, prior work suggests that habituation over time could affect an end-user's security and privacy behaviors [50].

## 7 CONCLUSION

The immersive nature of VR leaves users vulnerable to surveillance by bystanders or threats to physical safety. To alleviate that, users can use external cameras to view their surroundings. However, this might infringe on bystanders' privacy. In this work, we developed *Vice VRsa*, a novel concept and set of interactions to mediate the privacy of VR users and their bystanders. *Vice VRsa* transparently communicates presence, recording, and desired privacy through techniques inside VR (Halo, Radar, Live View) as well as to the bystander (Color Indicator and Public Display). In our preliminary evaluation with VR experts, we found that *Vice VRsa* could help address privacy concerns for both, VR user and bystander. We see *Vice VRsa* as an initial step towards addressing the emerging problem regarding mutual awareness between VR users and bystanders about their privacy in VR hybrid settings.

## ACKNOWLEDGMENTS

We would like to thank all the members of the HCI & Visualization research group at Autodesk Research for their help and support in the ideation and development of our project. We are also grateful to our reviewers for their valuable reviews in helping polish and strengthen our manuscript.

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
