# OpenReview forum: "Vice VRsa: Balancing Bystander's and VR user’s Privacy through Awareness Cues Inside and Outside VR"
_graphicsinterface.org/Graphics_Interface/2023/Conference_SD — GI 2023 - second deadline_

### Official Review · Reviewer_9CTJ · 2023-04-22
**I like that the design is bi-directional, providing ways of negotiation**

**Rating:** 7
**Confidence:** 4

**Review:**

This work introduces VICE VRSA, a framework designed for both VR users and bystanders: 1) the framework includes a VR interface that allows users to modify privacy modes and informs them of the existence of bystanders and 2) there is a public display + LED vest projector setup that conveys the current modes and data collection details to bystanders. A detailed, categorized expert evaluation is also provided (although the evaluation is conducted via Zoom and pre-recorded demo videos).

I like that the design is bi-directional, providing ways of negotiation. The visual designs signifying the user's privacy intent and warnings of privacy intrusion speak for themselves and do not require extra communication effort between the two parties, except for manual privacy mode selection. The literature review and evaluation section also aligns well with the major theme of the presentation.

While the use of existing classical designs, such as the Halo, Radar, and see-through views, did weaken the overall research contribution, I still consider this work valuable given that it proposes a privacy level design hosting these classical visualizations in a logical, reasonable way. (However, I do think that the orange/yellow colors are a bit confusing and need reconsideration).

The presentation quality is good, and Figure 3 clearly illustrates the VICE VRSA interactions.

In summary, I argue for a 'weak' acceptance of this work (7/10).

---

### Official Review · Reviewer_MmJz · 2023-04-23
**Limited study presentation but provided enough contributions**

**Rating:** 6
**Confidence:** 3

**Review:**

In this submission, the authors explored a way to balance the privacy between a VR users and bystanders.

Overall, I find this paper well-structured and clearly presented. It is easy to follow, and I do not have any difficulties understanding the core parts of this work.

Comments:
- The study part is an important drawback of this submission. The authors did not specify what the study design and procedure were. If the paper finally gets accepted, I would still argue that the authors should make efforts on describing the study.

 - For the halo inside VR, it is not clear why distinguishing by left/right is chosen. Is knowing left/right more important than knowing front/back from the privacy perspective? The authors may consider adding one/two sentences to explain this design choice. It would also be nice to provide how close/medium/far can be defined (like what distance can be considered as far).
 - For the color indicator outside VR, in my opinion, it needs some training for bystanders to understand the semantics (privacy levels in this work) behind the color representation. It might be good to also integrate the same color on the screen display, or add a bit of clue instead of only projecting the color.
- In the discussion, the authors claimed that "we explored the design space of balancing ...". I am not convinced that the design space is really explored (like what are the possible design options to make the balance instead of the implemented one).

Tu sum up, even though I am not happy with the study description that makes it hard to judge its validity, I think other parts should have already given enough contribution to the community, thus it may have a chance to be accepted.

---

### Official Review · Reviewer_z9QA · 2023-04-26
**Interesting work, need some clarifications**

**Rating:** 7
**Confidence:** 4

**Review:**

This paper presents a prototype system to explore visual cues that communicate VR usage patterns to outside bystanders. The system uses a combination of coloured lights and external display content to show the VR user's usage status as well as their interruption preference. The system also provides the VR user with information about people in their immediate surroundings and informs bystanders when they are being observed.

The concept is quite interesting although deserves some further thought. It is not completely clear where the main contribution of the paper is. The paper points to previous work that provides awareness to VR users of people nearby, however such work is not discussed in the related work section. The introduction states that the main contribution is a framework, however, there is no formal framework presented in the paper. All of this needs to be addressed and clarified.

The prototype and demonstrated examples are interesting but the presentation of this could be improved somewhat. Figure 3 is a bit confusing, particularly on the right side as all of the images are cropped very tightly and provide no context. There is no technical description of the physical prototype (camera and coloured lights). It's clear from the video that the floor projection is from an external source, however this is not mentioned in the paper.

The scenarios serve well in describing the basic concepts however in reality the usage would be somewhat more complex. For instance, it's not clear whether bystanders would be able to correctly interpret the meaning of the colours. They may also not behave so nicely, and for instance, some users may be opportunistic and try take advantage of the red privacy mode for intentional snooping. There is also no reason to expect that they would (or should) obey the request to leave the area in many contexts. It would be nice to see a discussion of these more complex scenarios and an acknowledgement that real world usage would be much more nuanced than in the examples.